# Effects of Antiangiogenetic Drugs on Microcirculation and Macrocirculation in Patients with Advanced-Stage Renal Cancer

**DOI:** 10.3390/cancers11010030

**Published:** 2018-12-28

**Authors:** Andrea Dalbeni, Chiara Ciccarese, Michele Bevilacqua, Marco Benati, Cristian Caimmi, Luca Cerrito, Federico Famà, Roberto Iacovelli, Anna Mantovani, Francesco Massari Alessandra Meneguzzi, Pietro Minuz, Martina Montagnana, Giovanni Orsolini, Maurizio Rossini, Gianpaolo Tortora, Ombretta Viapiana, Cristiano Fava

**Affiliations:** 1Department of Medicine, General Medicine and Hypertension Unit, University of Verona, 37126 Verona, Italy; marco.benati@univr.it (M.B.); luca.felice.cerrito@gmail.com (L.C.); fede.fama@gmail.com (F.F.); annamantovani4@gmail.com (A.M.); pietro.minuz@univr.it (P.M.); cristiano.fava@univr.it (C.F.); 2Department of Medicine, Oncology Unit, University of Verona, 37126 Verona, Italy; ciccarese.c@gmail.com (C.C.); roberto.iacovelli@univr.it (R.I.); francesco.massari@univr.it (F.M.); giampaolo.tortora@univr.it (G.T.); 3Department of Medicine, Oncology Unit, Fondazione Policlinico Universitario “Agostino Gemelli” IRCCS, 00168 Rome, Italy; 4Department of neurosciences, biomedicine and movement sciences, Section of Clinical Biochemistry, University of Verona, 37126 Verona, Italy; marco.benati@univr.it (M.B.); martina.montagnana@univr.it (M.M.); 5Department of Medicine, Rheumatology Unit, University of Verona, 37126 Verona, Italy; cristian.caimmi@univr.it (C.C.); giovanni.orsolini@univr.it (G.O.); maurizio.rossini@univr.it (M.R.); ombretta.viapiana@univr.it (O.V.); 6Department of Medicine, Section of Internal Medicine, University Laboratory for Medical Research (LURM) and Regional Centre for the Study of Platelets, University of Verona, 37126 Verona, Italy; alessandra.meneguzzi@univr.it

**Keywords:** TKI, angiogenesis, VEGF, renal cancer, NO (Nitric Oxide), endothelin-1, capillaroscopy, hypertension

## Abstract

Adverse cardiovascular effects, including hypertension, were described in patients with different cancers treated with tyrosine kinase inhibitors (TKI). The mechanism of TKI-related hypertension is still debated. The aim of this work was to study the effects of TKI on blood pressure (BP), searching for a relationship with possible causative factors in patients with metastatic renal cell carcinoma. We included 29 patients in a prospective, observational study; 22 were treated with a first-line drug (sunitinib), while seven participated in the second-line treatment (axitinib or cabozantinib). Patients were investigated at the beginning of antiangiogenic therapy (T0) and at one (T1), three (T2), and six months (T3) after treatment. Patients were evaluated by office blood pressure (BP) and ultrasonography to measure flow-mediated dilatation (FMD), and carotid artery distensibility (cDC) by echocardiography and nailfold capillaroscopy. Plasma endothelin-1 (p-ET-1), urine nitrates, and proteins were also measured. At T1, systolic BP, along with U proteins and p-ET-1, increased significantly. In patients with a clinically significant increase in BP (defined as either the need for an antihypertensive drug or systolic blood pressure (SBP) T1–T0 ≥10 and/or SBP ≥140 mmHg and/or diastolic blood pressure (DBP) T1–T0 ≥5 and/or DBP ≥90 mmHg), the urine nitrate concentration was lower at T0, whereas there were no differences in the p-ET-1 and U proteins. Seventeen participants showed changes in the capillaroscopic pattern at T1 with no association with BP increases. There were no differences in the FMD, cDC, and echocardiographic parameters. Our findings are consistent with those of previous studies about BP increases by TKI, and suggest a role of nitric oxide in BP maintenance in this population.

## 1. Introduction

In the last several years, antiangiogenetic therapy was included in the therapeutic schemes of many oncological diseases, including renal cancer [1]. In particular, tyrosine kinase inhibitors have been approved for the treatment of metastatic renal cell carcinoma [2]. Their biological activity interferes with tumor angiogenesis by blocking the intracellular signaling cascades of various soluble molecules that are involved in the growth of new blood vessels, such as VEGF (Vascular Endothelial Growth Factor), PDGF (Platelet-Derived Growth Factor), c-Kit-ligand, and FLT-3 (Fms Related Tyrosine Kinase 3) [1]. Despite their tolerability, especially compared with that of classic chemotherapy, these new drugs have several adverse effects. Angiogenesis inhibitor-related hypertension is one of the most common features of their cardiovascular toxicity [3]; moreover, the appearance of proteinuria or the deterioration of a pre-existing condition is also very frequent [4]. Hypertension of any grade according to CTCAE (Common Terminology Criteria for Adverse Events) is reported with an incidence ranging from 17% to 49.6% [4] depending on different TKI types, dosages, genetic polymorphisms of *VEGF* and *VEGFR*, age, body mass index (BMI), and pre-existing hypertension [5]. The increase in blood pressure (BP) occurs rapidly, generally within a few hours of the beginning of therapy [3,6], and appears reversible upon the discontinuation of antiangiogenic drugs. The overall incidence of proteinuria is reported to be from 8% to 20.7%, but only 1% of patients develop grade-3 proteinuria or nephrotic syndrome [4]. 

The pathogenetic mechanisms of antiangiogenic tyrosine kinase inhibitor (TKI)-related hypertension are not completely understood, and there are probably multiple mechanisms of action [3,7,8]. The interaction between VEGF and its receptor leads to a lower induction of NOsynthases in the endothelium; during sunitinib administration, urinary nitrates levels decreased significantly both in rats and humans [8,9], Furthermore, VEGF pattern blockade is associated with increased oxidative stress and reduced NO bioavailability [7]. It was noted that the increase in BP in mice treated with VEGFR2 inhibitors was prevented by the administration of the endothelial isoform of nitric oxide synthase (eNOS) inhibitor N(w)-nitro-L-arginine methyl ester (L-NAME) [8]. Another study reported that patients with sunitinib-related hypertension showed a twofold increase in plasma endothelin-1 (p-ET-1) [9]; intriguingly, this hypertensive effect did not occur when it was associated with the endothelin-1 (ET-1) receptor inhibitor in rats [10]. Microvascular rarefaction associated with anomalies of capillary morphology represents another mechanism proposed for TKI-related hypertension [11,12], which is mainly due to the reduction of the vascular network and an increase in the peripheral resistance. Indeed, renal damage, salt sensitivity, and proteinuria could be implicated [10]; indeed, in a clinical trial, a significant increase in plasma creatinine was shown in 5.2% of patients treated with sunitinib [4]. Other authors have suggested that a greater arterial stiffness could also be a factor in the vascular remodeling induced by antiangiogenic drugs [12]. The aim of the study was to assess which of the above-mentioned factors could be implicated in the rise of BP that occurs after TKI therapy for metastatic renal cell carcinoma. 

## 2. Methods

### 2.1. Study Design

This study was approved by the Ethical Committee of Verona (CESC n. 281). All of the subjects signed written informed consent.

The patients that were involved in this single-center, prospective, observational study were selected from the Oncology Department of the Azienda Ospedaliera Universitaria Integrata (AOUI) in Verona. The inclusion criteria were as follows: a diagnosis of metastatic renal cell carcinoma (histological diagnosis) with indication to start TKIs (such as sunitinib, pazopanib, axitinib, or cabozantinib), an age of 18–90 years, and a normotensive or hypertensive classification with a well-controlled BP (<140/90 mmHg). We excluded subjects with hepatic disease or severe hypertransaminasemia, chronic renal failure (plasma-creatinine >1.5 mg/dL), pregnancy, a history of abuse of alcohol or drugs in the last six months, or an inability to cooperate or undergo outpatient examinations. After verifying the inclusion/exclusion criteria, the patients were sent to the outpatient clinic of the General Internal Medicine and Hypertension for vascular exams (see below). The patients were evaluated before the start of the antiangiogenetic therapy (T0) and at one month (T1), three months (T2), and six months (T3), after the initiation of TKIs. Antihypertensive therapy was modified by our team at each study point according to the clinical need.

On the visit day, after a brief interview, subjects underwent BP measurement, endothelial function assessment by ultrasounds, capillaroscopy, echocardiography, and the collection of venous blood and extemporary urine samples. Blood was collected in tubes anticoagulated with EDTA (ethylendiaminetetracetic acid) and in vacuum tubes containing no additives (Vacutest Kima, Kima, Arzergrande, Padova, Italy). Urine was stored at −20 ° C until use, and then thawed at 4 °C and centrifuged at 2400 *g* for 10 min. Plasma and/or serum were frozen at −80 °C until analysis.

Tumor assessments (by RECIST, version 4.0) were performed every 12 weeks from the start of treatment. 

### 2.2. Blood Pressure Measurements

BP was measured with an oscillometric device (TM-2501, A&D instruments Ltd., Abingdon Oxford, UK) in a clinostatic position at rest; for all of the patients, we averaged three BP measurements performed five min apart. The patients were also invited to check their BP regularly at home. Since these home blood pressure measurements were not taken using the same device, but each patient used its own, these data were intended for clinical use, and are not presented. 

### 2.3. Laboratory Plasma, Serum, and Urine Analysis

Urinary nitrates were measured using an immunoenzymatic colorimetric method through a specific laboratory kit (Caymann Chemical Nitrate/Nitrite Assay Kit, Cayman Chemical, Ann Arbor, MI, USA). The assay was performed according to a procedure recommended by the manufacturer. The intra–interassay coefficients of variation were 2.7% and 3.4%, respectively. The results are expressed as a nitrate/creatinine ratio (µM/µmol). After centrifugation at 1500 *g* for 10 min at room temperature, the serum was separated, stored in aliquots, and kept frozen at −80 °C until measurement.

Serum ET-1 levels were measured using the Endothelin-1 Quantikine ELISA Kit (R&D Systems, Inc., Minneapolis, MN, USA), according to manufacturer’s instructions. All of the samples were analyzed in duplicate. The limit of detection of this method is 0.207 pg/mL, as stated by the manufacturer. The reported intra-assay and interassay precision were <4% and <8%, respectively. A standard curve was generated by plotting the absorbance versus the log concentration using a four-parameter logistic curve fit. The urinary protein–creatinine ratio (mg/mmol creatinine) was measured in spot urine samples. Analysis was performed by an immunoturbidimetric assay on a Roche Diagnostics Cobas 8000 analyzer (Roche Diagnostics, Mannheim, Germany).

### 2.4. Evaluation of Vascular Parameters (FMD and Carotid Distensibility)

The detection of flow-mediated dilatation (FMD) (a surrogate for endothelial functionality) was performed in a fasting state according to international guidelines [13]: briefly, a high-resolution Doppler ultrasound (LogiQ P5 pro, GE Healthcare, Indianapolis, IN, USA) equipped with a five to 13-Hz probe and 0.01-mm axial resolution was used to evaluate the diameter and the flow of the brachial artery and its walls (in particular, the intima-lumen interface). Thanks to dedicated hardware ("FMD study", Quipu, Pisa, Italy) that can detect multiple images per second (more than 25), we could measure the arterial diameters and the percentage of variation of the diameter after cuff deflation based on the flow-mediated dilation guidelines. Then, we measured the carotid intima-media thickness (cIMT) within one centimeter from the carotid bulb using specific hardware (Carotid Studio, Quipu, Pisa, Italy). Finally, the common carotid artery distensibility (CD) was calculated as: CD = ΔA/(A* ΔP) where A is the diastolic lumen area, ΔA is the stroke change in the lumen area, and ΔP is the pulse pressure (PP). Changes in diameters were detected using ultrasound B-mode image sequences of the right and left common carotid arteries that were acquired at different steps and analyzed by the above-mentioned automatic system. 

### 2.5. Echocardiography

The main morpho-functional cardiac parameters were measured in patients by echocardiography. In particular, we recorded the thickness of the interventricular septum (considering a limit of 10 mm as a cut-off for the diagnosis of cardiac hypertrophy), the main left/right diameters and volumes, the ejection fraction, and the E/A ratio (a diastolic dysfunction index when greater than 1).

### 2.6. Capillaroscopy

A periungual capillaroscopic evaluation was performed on all of the patients at each of the four study points (T0, T1, T2, and T3) by a single independent expert rheumatologist blinded for information about the patients’ history or previous capillaroscopies. An OPVC (optical probe videocapillaroscopy) with polarized light and variable magnification was used. The instrumentation consisted of a PAL (phase alternating line) color camera with magnification from 100× to 1000×, a source of incident, cold, and monochromatic light on the field, and a personal computer (PC) and monitor on which to view and save images, even comparing them with subsequent assessments. The modifications of the capillary pattern were classified using a semi-quantitative method in three categories (absent, mild–moderate, and marked changes) because of a change in capillary density that was not clearly detectable.

### 2.7. Radiological Imaging of the Disease

CT images were evaluated jointly by oncologists and radiologists, according to the RECIST (Response evaluation criteria in solid tumors) criteria v. 4 (standard criteria for the radiological evaluation of the effect of chemotherapy in antineoplastic therapy). The appearance of new focal lesions or the enlargement of pre-existing lesions were considered signs of disease progression (with an increase of at least 20% of the transverse diameters); stable lesions were defined as having dimensional changes of up to ± 20%, while a partial response was defined as a reduction greater than or equal to 30%, which was also classified as regression. 

### 2.8. Statistical Analysis

Continuous variables are expressed as the mean ± standard deviation (SD). The comparison between continuous variables measured at different study points in each participant was performed using the Student’s T-test for paired data and/or the Wilcoxon rank test, depending on the normal distribution of the analyzed variables. Comparison between the continuous variables among the patients was performed with the Student’s T-test for unpaired data and/or the Mann–Whitney U test. To compare dichotomous variables, Fisher’s exact test was used, while the correlations were completed with the Pearson test or Spearman test. The Kaplan–Meier estimator was used to obtain progression-free survival (PFS) and overall survival (OS); a two-sided 95% CIs Cox proportional hazards model was performed for the medians and hazard ratios (HRs) for each end point. In all of the cases, a statistically significant value was considered with at *p* < 0.05. SPSS Statistics 22 (IBM SPSS Statistics, Ltd, Rm 1804, 18/F, Westlands Centre, Westlands Road, Quarry Bay, Hong Kong) and GraphPad Prism 7 (GraphPad Software, 2365 Northside Dr. Suite 560 San Diego, CA, USA) were used for all of the data analysis.

## 3. Results

### 3.1. Baseline Population Characteristics 

Twenty-nine patients participated in this study. Twenty-two patients were treated with sunitinib, whereas three and four participants were treated with axitinib and cabozantinib (second-line or third-line therapy), respectively. Since the mechanisms of TKIs are similar, their effect on BP is strictly linked to the assumption of the drugs, and is due to substantial pharmacological wash-out (at least two months between the first and successive lines). For this reason, we pooled all of the patients’ data, regardless of the line of therapy in the main analysis. Nevertheless, all of the analyses were repeated in the subgroup of 22 sunitinib-receiving patients (first-line therapy). The sample included five females and 24 males, aged between 45–76 years (64.7 ± 9.0 y) whose average BMI was 25.7 ± 4.5; 65% and 58.6% of them were prior smokers or hyperlipidemic, respectively. All of the subjects were previously treated locally by surgical intervention (partial or total nephrectomy and/or lymphadenectomy, adrenalectomy, and perirenal fat resection) or systemically (surgery of oligometastatic disease) and had indication of medical therapy with antiangiogenic drugs. The most common histology (*n* = 27) was clear cell renal cell carcinoma (ccRCC), and only a few tumors (*n* = 6) were non-clear cell RCC (nccRCC). Based on the international metastatic RCC database consortium (IMDC) prognostic classification [14], 39.4% of the entire population had a good prognosis, 39.4% had an intermediate prognosis, and 12% had a poor prognosis.

Twenty-one patients completed all of the phases of the protocol, four completed T0–T1 and T2, and four completed only the first two steps (T0–T1). The causes of drop-out were either a severe worsening of performance status or death. At baseline, 14 subjects (48.3%) underwent antihypertensive therapy (Angiotensin-Converting enzyme inhibitor, angiotensin II receptor blockers, calcium-channel blocker, diuretics, β-blockers). At T1, the introduction of antihypertensive drugs was needed in four patients (two treated with calcium-channel blockers only, and two treated with a combination of an ACE inhibitor and calcium-channel blocker); a second antihypertensive drug was added for seven of the 14 subjects who were previously under treatment. At T2, three patients started calcium-channel blocker monotherapy, while a third drug was added for two subjects with prior dual medical therapy. In summary, 18 of 29 (62.1%) and 17 of 25 (86.2%) participants were treated with antihypertensive drugs at T2 and T3, respectively. 

### 3.2. Blood Pressure Increase

After one month of TKIs and without any changes in the antihypertensive therapy, the average office systolic BP, but not the diastolic BP, increased significantly compared to that at baseline (see Table 1). At T2 and T3, after the above-mentioned addition of antihypertensive drugs, both the systolic and diastolic BPs remained unchanged from that at baseline (Table 1). Thus, when we compared the systolic and diastolic BP (SBP and DBP, respectively) trends between baseline (T0) and first follow-up (T1), we could clearly identify two subgroups according to what we considered clinically significant BP increases (>10 mmHg in SBP and/or >five mmHg in DBP or a final BP ≥140/90 mmHg or the need for an antihypertensive drug). We labeled this subgroup the “TKI-sensitive” group to distinguish it from the “TKI-insensitive” group, comprising patients who neither underwent significant BP alterations nor required any antihypertensive drugs at T1. Fourteen of 29 (48.3%) and 11 of 22 subjects (50.0%) showed a clinically significant increase in BP (that is, they were considered “TKI-sensitive”) in the whole sample and in the sunitinib subgroup, respectively. 

TKI-sensitive and TKI-insensitive participants did not show any significant difference regarding their anthropometric and anamnestic characteristics (Appendix A). Both systolic and diastolic BP were similar at baseline and thereafter in patients taking antihypertensive therapy or not (Appendix A). 

### 3.3. Urinary Nitrates

A significant reduction in urinary nitrates at T3 was found in the whole sample compared to that at baseline (Table 1). Baseline nitrates were lower in the subgroup of TKI-sensitive patients (Table 2, Figure 1). The same results were obtained when analyzing only the 22 sunitinib-treated patients (nitrate/creatinine 53.11 ± 21.17 μM/μmol in TKI-sensitive patients compared to 120.09 ± 94.3 μM/μmol in TKI-insensitive patients, *p* = 0.032). Furthermore, basal urinary nitrates were significantly lower in subjects who were already taking antihypertensive therapy at baseline (Appendix A).

### 3.4. Proteinuria and ET-1

Considering the whole population, both U proteins and p-endothelin-1 increased significantly between baseline and T1 (U proteins = 20.7 ± 27.67 mg/mmol at T0 versus 40.03 ± 60.12 mg/mmol at T1, *p* = 0.023; *p* = ET-1 = 3.35 ± 0.84 pg/mL at T0 versus 5.00 ± 1.85 pg/mL at T1, *p* < 0.001; Figure 2 and Figure 3). We did not observe any differences in both proteinuria and ET-1 at baseline and at T1 in the two subgroups identified according to either the clinically significant BP increases (Figure 2 and Figure 3) or the use of antihypertensive therapy at baseline (Appendix A). The same results were confirmed in the sunitinib-treated subpopulation. 

### 3.5. Capillaroscopy

At baseline, no patient exhibited relevant morphological anomalies of periungual microcirculation. A mild–moderate modification of the capillary pattern, but not in capillary density (especially tortuosity, reduction of capillary diameter and global architectural disorder), was noticed in 17 of 29 subjects (58.6%) at T1 (Figure 4). At T2 and T3, the percentage of mild–moderate capillaroscopic anomalies increased to 75.0% (18/24) and 83.8% (15/18), respectively. At T2 and T3, five and six subjects of those with previous capillary changes showed a further deterioration on imaging, respectively (marked changes). Capillaroscopic changes were not significantly different in the TKI-sensitive patients compared to those in the TKI-insensitive subjects at one, three, and six months of follow-up.

### 3.6. Vascular Parameters and Echocardiography

No significant changes in FMD, carotid distensibility, cIMT, and the tested ultrasound cardiac parameters, which were compared at different study points, were detectable in the whole sample or in the sunitinib subgroup (Table 1). Even when dividing the sample into TKI-sensitive and TKI-insensitive patients, no differences between these measures were found (Table 2, Appendix A).

### 3.7. Evaluation of Disease Wuìith CT Imaging

Antiangiogenic therapy was shown to be significantly effective in shrinking peripheral metastasis in more than half of the participants (Figure 5) or stop disease progression in the remaining ones. Among patients treated as first-line therapy, 56% were progression-free at 12 months (median progression-free survival (PFS) 10.8 months, Figure 6a). For patients treated with the second or third-line therapies, the progression-free survival (PFS) rate at six months was 50%. The overall survival (OS) rate of patients treated with first-line therapies at 24 months was approximately 60% (median OS 28.8 months, Figure 6b), while in patients treated with second or third-line therapies, the OS rate at 12 months was 40%. These data are in line with those reported in the literature [15,16]. We did not find a clear correlation between either PFS or OS with TKI sensitivity.

## 4. Discussion

In this study, we confirmed that a high percentage of patients treated with TKI have a significant increase in BP. We also showed that BP trajectories can be markedly different in different patients. In particular, some subjects showed a marked increase of BP values within the first month of TKI therapy, whereas others showed stable values. We considered a BP increase to be clinically significant if there was both the need to introduce an antihypertensive drug and if the final BP was above 140/90 mmHg, or when an increase greater than 10 mmHg in SBP or five mmHg in DBP occurred.

Our definition is somewhat arbitrary, but it allowed us to divide our sample not only according to hypertension status, but also into groups of patients who were sensitive or non-sensitive to TKIs. Since our main focus was pathophysiological rather than clinical, we chose this definition to allow a better distinction between factors associated with the increase in BP in some, but not all, the patients. Moreover, we wanted to show that the application of even more stringent cut-offs (BP increase: >20 mmHg in SBP and/or >10 mmHg in DBP) did not shift any patients from the subgroup that was initially identified to the opposite subgroup. 

It has been previously shown that an increase in BP appears quickly when TKI therapy is started, and returns to its previous values quite rapidly in most patients after the discontinuation of TKI [17]. Previous studies have reported that the prevalence of a BP increase in each grade in patients receiving "VEGF-signaling pathway" inhibitors, such as axitinib, sunitinib, pazopanib, and sorafenib, varies from 17% to 72%. Thus, our data are in line with those shown by other groups [18,19,20,21]; in fact, four weeks after the initiation of targeted therapy, we observed an increase in the BP in 14 of 29 patients (48.3%). Subsequently, the average BP recovered to values similar to those at baseline, which was probably due to the addition of antihypertensive drugs. Several studies have investigated the pathophysiological basis of TKI-related hypertension and showed differences in ET-1, proteinuria, and endothelial function [21]. However, these factors have seldom been analyzed together, and many doubts remain about the exact role of different factors. 

In our sample, we found a significant difference only in the basal urinary nitrates (a surrogate marker of both endogenous and dietary nitrate exposure) [22] between participants who showed significant increases in their BP and those with no significant modifications, while the differences between baseline and T1 was not significantly changed. This suggests a potential role of nitric oxide and the endothelium in this setting; we hypothesized that patients with a reduced “reserve” of nitric oxide could be more sensitive to TKI vascular side effects. It is known that VEGF inhibition is associated with lower genetic transcription and the enzymatic activation of eNOS, which is the endothelial isoform of nitric oxide synthase [10,23]. Some authors also suggested a reduced vasodilating activity due to an increase in oxidative stress and a consequently reduced bioavailability of NO itself [7]. In a recent study in mice, the urinary metabolites of plasma nitrates from mice treated with sunitinib were found to be dose-dependently decreased after eight days of administration, whereas they were increased, even if they did not reach baseline level, after drug withdrawal [6]. Furthermore, in a study involving 27 patients taking pazopanib, Tinning et al., hypothesized a consistent role of the reduced renal medullary formation of NO after detecting a reduction in the urinary metabolites of NO (NOx and cGMP, cyclic-Guanosine Monophosphate) [24]. A partial confirmation derives from the effective use of long-acting NO donor drugs in hypertensive patients on TKI [25] and their superiority compared to ACE inhibitors and calcium-channel blockers in patients who are resistant to conventional drugs [26]. 

Furthermore, a significant increase of plasma ET-1 was detected in our sample after four weeks of therapy, but no clear correlation with BP was found. These results seem to argue against a pivotal role of ET-1 in the observed increase of BP in TKI-sensitive patients, but could be better interpreted in light of the above-mentioned reduced functional “reserve” of NO. That is, even if ET-1 increases in all of the patients, the effects on BP occurs preferentially in patients with an impaired reserve of NO. 

ET-1 is a potent vasoconstrictor that is produced by the endothelium in response to various stimuli, which is also associated with renal damage and cardiovascular remodeling. The ET-1 system was found to be activated in sunitinib-treated rats [9]; its causal role in TKI-associated hypertension was proposed, since in rats treated with sunitinib, the increase in BP was prevented by Macitentant, which is an ET-1 receptor-specific blocker [10]. In our sample, a marked increase in ET-1 with respect to the baseline was found in almost all of the patients, regardless of whether or not a clinically significant rise in BP had occurred. Thus, other mechanisms also exist.

Renal injury and proteinuria induced by antiangiogenic TKI has also been proposed as a potential explanation for the onset of hypertension [21]. In a study involving 375 sunitinib-treated participants, Motzer et al., documented an all-grade increase in plasma creatinine in approximately 70% of subjects [27], while a median relative reduction of the estimated glomerular filtration rate (eGFR) of 25 ml/min was found in a cohort of sunitinib-receiving or sorafenib-receiving patients [28]. The glomerular and podocyte damage that were associated with vasoconstriction induced by the impaired activity of VEGF were proposed mechanisms; a probable contribution could be derived from previous nephrectomy and CT contrast-induced nephropathy [4]. Despite these premises, neither a statistically significant increase in plasma creatinine or eGFR was found in the whole sample or in subgroups. Regardless, the design of our study cannot exclude the possibility that renal failure could play a role in long-lasting hypertension, because an average time between initial TKI administration and decreases in eGFR was reported to be approximately 199 days [4], which is a longer period than that of our protocol. An increase in urine proteins after the first month of therapy was also detected in our sample, which is in line with many studies in which the incidence of proteinuria is reported to be between 5–73% [4]; the pathogenesis has been shown to be related to different factors, such as the type and dosage of antiangiogenic chemotherapy and certain pre-existing characteristics of the population. A pathogenetic role of the VEGF-signaling blockade in determining proteinuria has been proposed due to the importance in the maintenance of the glomerular filtration barrier. In fact, VEGF is essential to the promotion of podocyte homoeostasis, the upregulation of glomerular barrier proteins (podocin, CD2-associated protein, actin, and α- actinin-4) and the preservation of renal blood flow, as shown in the murine model knock-out for VEGF [29]. 

Hypertension itself has also been associated with proteinuria; however, proteinuria in hypertension usually develops slowly, and it is difficult to postulate that the rapid increase in BP after the antiangiogenic drugs could be the cause of proteinuria. In light of these considerations, although a direct role cannot be excluded, we hypothesized that proteinuria could be an epiphenomenon of TKI-induced endothelial dysfunction, which is a process that could explain both situations.

Regarding the effects on microcirculatory patterns induced by anti-VEGF TKI, an increasing number of participants developed a clinical and demonstrable change in microvessels at periungual capillaroscopy, especially in terms of tortuosity, reduction in vascular diameter, and architectural disorder, but without a clear decrease in capillary density. In fact, capillary rarefaction has been demonstrated in the oral mucosa of sunitinib and telatinib-treated patients [11,30]; its putative relationship with the development of hypertension may also due to a possible contribution of small vessels in determining systemic vascular resistance. Furthermore, we did not see a statistically significant diversity in term of capillaroscopic findings in the two subgroups that were defined according the presence of BP increases after one month of therapy. For proteinuria, we cannot exclude a role of vascular remodeling in hypertensive patients with a long history of anti-VEGF assumption; however, our data do not support a major role in the immediate rise of BP after TKI initiation.

Despite differences in the basal nitrates between hypertensive and stable subjects, our data about endothelial function when measured non-invasively by the FMD technique were not different according to the BP rise both at baseline and at different study time points. It is not clear why endothelial function was apparently not modified in line with urinary nitrates, but it should be emphasized that FMD depends 30–50% on the release of NO by the endothelium [31,32], and that urinary nitrates could reflect the kidney production of NO more than systemic production.

Finally, we did not find a significant difference in the local distensibility of the carotid artery by ultrasound measurement between the different time points and the two subgroups. Thus, the hypothesis of a role of anti-VEGF drugs serving as co-factors in determining hypertension through an increase of arterial stiffness [33], at least in the short term, is not corroborated by our data. 

Our study has several limitations, including the small sample size and the difficulty in interpreting functional data following the use of antihypertensive medications. Furthermore, different anti-VEGF drugs were used in different lines of therapy and in various histological types of renal cancers, and this could have affected some measurement. However, it still provides important insights into the possible interpretation and the mechanisms involved in the BP rise that occur after the initiation of TKI therapy, indicating a pivotal role for NO in the endothelin system.

## 5. Conclusions

In conclusion, in a group of patients with renal cell carcinoma, we have shown that the effects of antiangiogenic TKI on BP are detectable in nearly 50% of cases, and that urinary proteins and p-ET-1 rise in almost all patients in less than one month. The TKI-sensitive patients had lower urine nitrates than TKI-insensitive patients at baseline, but no specific correlation was detectable between BP values and u-nitrates or all of the other tested factors. We observed changes in some endothelial-related factors, such as u-nitrates, p-ET-1, and proteinuria along with BP. This result suggests that the endothelium is the main culprit of these conditions, further indicating that factors or drugs that restore NO and other factors could be potentially useful to treat or prevent increases in BP. Further studies are needed to produce a better understanding of the relationships between all these factors, and determine whether some endothelial markers, such as u-nitrates, can be potentially useful for screening, as well as which can provide the best antihypertensive therapy for this condition.

## Figures and Tables

**Figure 1 cancers-11-00030-f001:**
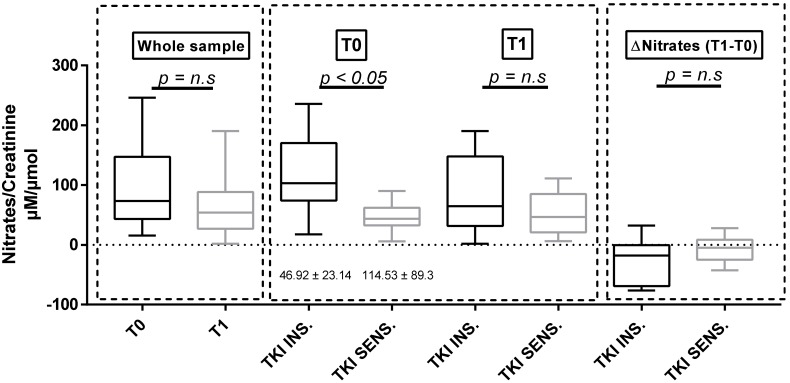
Nitrate/creatinine ratio at baseline and at T1 in the whole sample and in the two subgroups defined according to the sensitivity of BP to TKI therapy. TKI SENS = TKI-sensitive patients; TKI INS = TKI-insensitive patients, BP: blood pressure, n.s: no significant.

**Figure 2 cancers-11-00030-f002:**
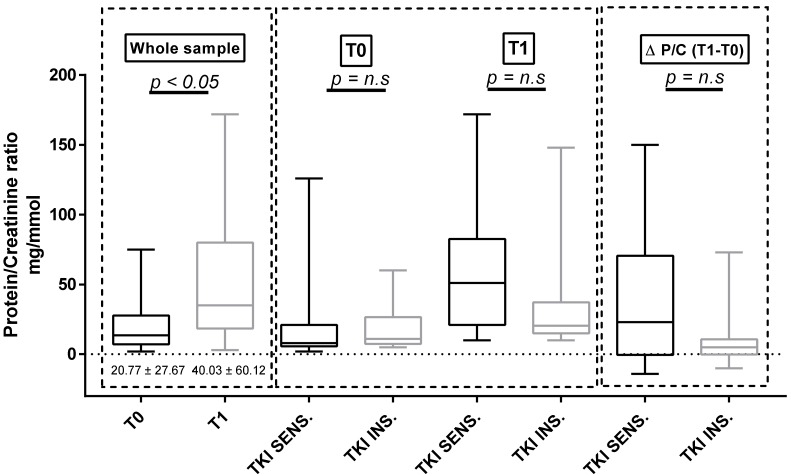
Protein/creatinine ratio at baseline and at T1 in the whole sample and in the two subgroups defined according to the sensitivity of BP to TKI therapy. n.s: no significant.

**Figure 3 cancers-11-00030-f003:**
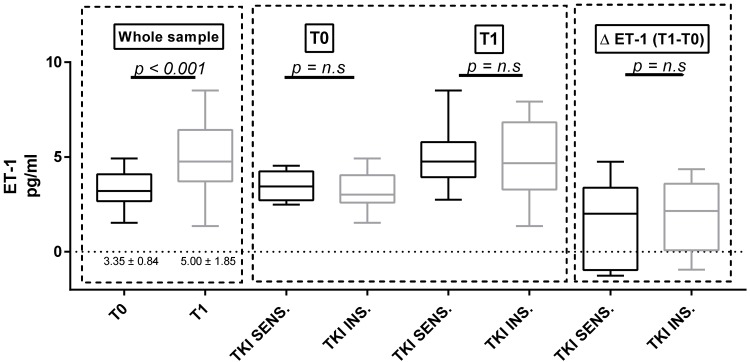
Endothelin-1 (ET-1) at baseline and at T1 in the whole sample and in the two subgroups defined according to the sensitivity of BP to TKI therapy n.s: no significant.

**Figure 4 cancers-11-00030-f004:**
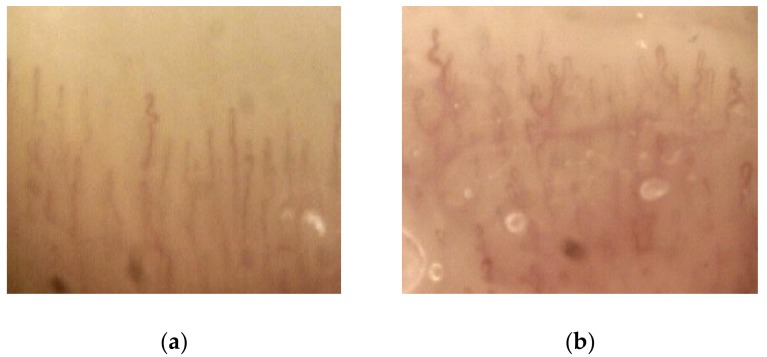
Representative capillaroscopic images at the beginning of antiangiogenic therapy (**a**) and at T1 (**b**): tortuosity and architectural disorder are shown after one month of therapy (magnification: 100× to 1000×)

**Figure 5 cancers-11-00030-f005:**
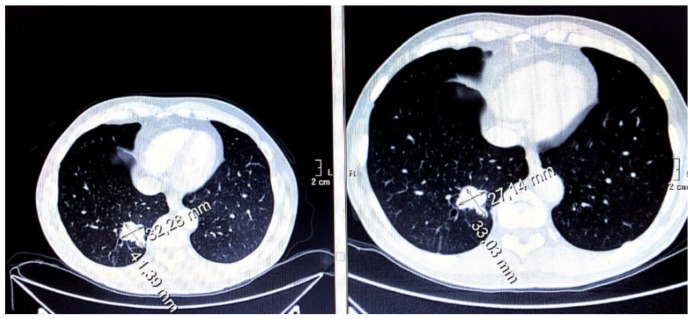
Effects of antiangiogenic therapy in terms of moderate dimensional reduction of pulmonary metastasis (in patients from December 2016 to March 2017).

**Figure 6 cancers-11-00030-f006:**
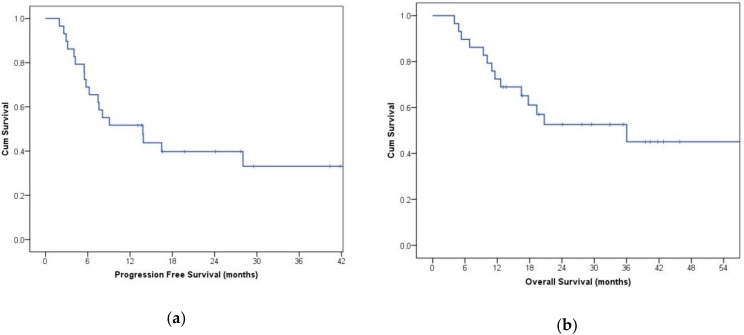
Kaplan–Meier curve of progression-free survival (PFS) (**a**) and overall survival (OS) (**b**) in participants treated with first-line therapy.

**Table 1 cancers-11-00030-t001:** Main cardiovascular and laboratory parameters in the whole sample.

Parameters	Baseline (T0, *n* = 29)	1 Month (T1, *n* = 29)	3 Months (T2, *n* = 25)	6 Months (T3, *n* = 21)
SBP (mmHg)	136.5 ± 14.9	145.7 ± 17.3 ^†^	137.4 ± 11.4	133.7 ± 15.1
DBP (mmHg)	85.3 ± 10.	87.9 ± 11.3	85.7 ± 8.3	80.7 ± 11.7
Creatinine (mg/dl)	1.21 ± 0.25	1.27 ± 0.28	1.17 ± 0.30	1.17 ± 0.30
GFR (mL/min)	66.06 ± 24.81	69.28 ± 26.32	63.62 ± 28.39	68.73 ± 28.23
Nitrates/Creatinine (µM/µmol of creatinine)	84.21 ± 73.43	70.57 ± 67.84	73.95 ± 75.55	72.31 ± 52.33 ^†^
Platelets (10^3^/µL)	257.4 ± 103.0	190.2 ± 76.4 ^‡^	226.8 ± 129.9	225.6 ± 94.3
IMT (mm)	0.78 ± 0.20	0.73 ± 0.12	0.75 ± 0.14	0.70 ± 0.15
CD (10^3^ kpa)	19.47 ± 9.10	20.71 ± 15.86	19.42 ± 8.10	17.83 ± 7.73
FMD (%)	3.75 ± 5.95	3.43 ± 5.24	3.15 ± 3.73	3.33 ± 6.46
IVS (mm)	0.97 ± 0.15	0.99 ± 0.21	0.96 ± 0.15	0.95 ± 0.19
Indexed LVM (LVM/BSA. g/m^2^)	99.78 ± 78.66	93.26 ± 44.73	97.38 ± 44.73	85.18 ± 33.24
EF (%)	58.90 ± 7.51	56.94 ± 7.37	56.11 ± 8.60	57.63 ± 7.67
E/A	0.74 ± 0.19	1.07 ± 0.62	0.94 ± 0.55	0.87 ± 0.32

SBP, systolic blood pressure; DBP, diastolic blood pressure; GFR, glomerular filtration rate; IMT, intima-media thickness; CD, carotid distensibility; FMD, flow-mediated dilatation; IVS, inter-ventricular septum; LVM, left ventricular mass; BSA, body surface area; EF, ejection fraction; E/A, wave E and A ratio; **^†^***p* < 0.05 compared to baseline; **^‡^***p* < 0.001 compared to baseline.

**Table 2 cancers-11-00030-t002:** Comparison of the main cardiovascular and laboratory parameters in the two subgroups defined according to the occurrence of a clinically significant increase in blood pressure (BP) after tyrosine kinase inhibitor (TKI) initiation.

Parameters	TKI-Sensitive (14/29)	TKI-Insensitive (15/29)	TKI-Sensitive (14/29)	TKI-Insensitive (15/29)	TKI-Sensitive (11/25)	TKI-Insensitive (14/25)	TKI-Sensitive (8/21)	TKI-Insensitive (13/21)
	Baseline (T0, *n* = 29)		1 month (T1, *n* = 29)		3 months (T2 *n* = 25)		6 months (T3, *n* = 21)	
SBP (mmHg)	129.6 ± 12.8	142.1 ± 14.5 ^§^	154.9 ± 16.6	138.2 ± 14.2 ^†^	136.1 ± 15.5	138.3 ± 8.4	128.0 ± 21.0	136.6 ± 14.3
DBP (mmHg)	79.7 ± 7.6	89.9 ± 10.0 ^†^	93.4 ± 8.7	83.50 ± 11.4 ^§^	84.6 ± 9.2	86.4 ± 7.3	75.1 ± 12.9	83.4 ± 10.4
Platelets (10^3^/µL)	363.5 ± 227.3	217.6 ± 54.8 ^§^	236.9 ± 136.2	176.0 ± 57.1	260.1 ± 192.9	206.1 ± 68.1	234.5 ± 103.1	221.0 ± 93.2
Creatinine (mg/dL)	1.36 ± 0.63	1.23 ± 0.31	1.20 ± 0.74	1.29 ± 0.30	1.23 ± 0.30	1.13 ± 0.33	1.04 ± 0.26	1.23 ± 0.30
Nitrates/Creatinine(µM/µmol of creatinine)	46.92 ± 23.14	114.53 ± 89.3 ^§^	51.55 ± 36.49	86.02 ± 83.41	59.20 ± 90.80	84.77 ± 63.30	63.95 ± 41.13	75.89 ± 57.49
BMI (Kg/m^2^)	23.64 ± 3.23	27.28 ± 4.79 ^§^	24.09 ± 4.56	26.55 ± 4.26	23.64 ± 3.65	26.67 ± 4.59	24.25 ± 4.24	26.30 ± 4.67
CD (10^3^/KPa)	19.07 ± 10.63	19.80 ± 10.99	21.27 ± 22.14	20.26 ± 8.75	19.16 ± 7.90	19.60 ± 8.50	20.74 ± 11.16	16.37 ± 5.25
FMD (%)	2.88 ± 3.87	4.32 ± 7.08	4.09 ± 3.36	3.00 ± 6.27	2.61 ± 5.00	4.13 ± 5.16	1.05 ± 2.08	4.63 ± 7.15
cIMT (mm)	0.79 ± 0.21	0.77 ± 0.18	0.72 ± 0.13	0.74 ± 0.11	0.70 ± 0.14	0.79 ± 0.14	0.68 ± 0.16	0.71 ± 0.15
Indexed LVM (LVM/BSA g/m^2^)	81.04 ± 23.96	94.00 ± 47.38	96.02 ± 31.28	91.23 ± 53.50	98.59 ± 30.31	96.60 ± 64.98	86.47 ± 31.45	84.49 ± 35.23
EF (%)	58.81 ± 8.04	58.98 ± 7.32	57.85 ± 5.64	56.21 ± 8.64	59.08 ± 7.60	54.14 ± 8.91	57.84±5.68	57.52±8.70

TKI-sensitive, patients with a clinically significant increase in BP (see text); TKI-insensitive, patients without a clinically significant increase in BP (see text); SBP, systolic blood pressure; DBP, diastolic blood pressure CD, carotid distensibility; FMD, flow-mediated dilatation; medium cIMT, intima-media thickness (left and right at average); LVM, left ventricular mass; BSA, body surface area; EF, ejection fraction; *p*: § < 0.05, † < 0.001 (compared to TKI-sensitive patients at the same study points).

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
