# Peer review of "Effects of Antiangiogenetic Drugs on Microcirculation and Macrocirculation in Patients with Advanced-Stage Renal Cancer"

_cancers, 2018, doi:10.3390/cancers11010030_

Round 1

Reviewer 1 Report

The present study is investigating the potential causes of blood pressure rise happening in some patients with metastatic renal cell carcinoma and undergoing anti-angiogenic tyrosine kinase inhibitors treatment. While the study is of relatively small size, I believe the work is nicely written, the data are presented in a proper way, the discussion is well elaborated and the limitations are given making the work of interest for publication in Cancers Journal.

Author Response

Thank you very much for your comments and suggestions. 

Reviewer 2 Report

In this manuscript, Dalbeni et al, reported their findings about the factors associated with BP in patients  with RCC treated with TKI. They reported NO is associated with possible TKI induce HTN. Although this is small cohort of patient, it is worth to report since this is evolving area.

- There is minor typo thought out, please correct ( such as first paragraph of result)

- Patients started different anti-HTN drugs, this can also effect some of the measurements. Please include this in limitation.

- Discussion, paragraph 3, " change chemotherapy to targeted therapy since those agents are not chemo.

- Need to expend the limitations, different TKIs, different line of treatment, differences in histology, clear cell, non clear cell, some pts has history of baseline hypertension and on anti-hypertensive already, etc.

Author Response

Thank you very much for your comments and suggestions. 

The manuscript has been previously edited by expert reviewers (if requested the certification could be attached)

Sincerely, 

Michele Bevilacqua 

Reviewer 3 Report

In the current study, authors have done a great job trying to elucidate the possible pathogenetic mechanisms of a chemotherapy, antiangiogenic tyrosine-kinase inhibitor (TKI), related hypertension in a cohort of 29 metastatic renal cell carcinoma patients. They concluded that TKI therapy is related to a significant increase in blood pressure (BP), plasma ET-1 and proteinuria after 1 month treatment. The paper was very well organized and well written, easy to follow, and I believe the reader will benefit tremendously from this work. I recommend to accept after the following concerns are addressed.  

Major Concerns, 

1.    In Figure 1, 2, 3, besides the box plots shown in the figures, could the authors show the actual data points in each box plots, that would make the figure even more informative. 

2.    In the last section of the Results, that is titled “CT”. Authors were trying to show progression free survival (PFS) rate by the evidence of CT images. Besides the statistical analysis, could authors show representative CT images to support their points? Could authors show PSF and overall survival (OS) rate in kaplan meier curve? And, instead of “CT”, please use a more representative and descriptive title for this paragraph of the result.

3.    Since the work studies TKI on advanced renal carcinoma, if there is any, could authors show more results in TKI cancer treatment efficacy and also discuss more in the discussion. 

4.    Since the paper also studies weather TKI affects micro- and macro circulation, besides the statistical analysis, could authors show some representative images of periungual capillaroscopic evaluation and representative histogram of digital plethysmograph analysis.

Minor Concerns, 

1.    In Introduction, 5thline of 2ndparagraph, ‘’NO bioaviability[7].”, a potential typo, could it be “NO bioavailability”?

2.    In Introduction, 4thline from the bottom of the 2ndparagraph, “5,2% of patients”, could it be “5.2%”

Author Response

Dear reviewer, 

Thank you very much for your comments and suggestions

The manuscript has been previously evaluated by an expert language editor

As you suggested, I provide you representative CT and capillaroscopic images, Kaplan Meyer curves together with modified ghaphics

Sincerely, 

Michele Bevilacqua